



# Temperature seasonality in the North American continental interior during the early Eocene climatic optimum

Ethan G. Hyland[1,2*], Katharine W. Huntington[1], Nathan D. Sheldon[3], Tammo Reichgelt[4]

[1] Department of Earth & Space Sciences, University of Washington, Seattle, WA 98195

[2] Department of Marine, Earth & Atmospheric Sciences, North Carolina State University, Raleigh, NC 27695

[3] Department of Earth & Environmental Sciences, University of Michigan, Ann Arbor, MI 48104

[4] Lamont Doherty Earth Observatory, Columbia University, Palisades, NY 10964

*Correspondence to: Dr. Ethan G. Hyland (ehyland@ncsu.edu)*

**Abstract.** Paleogene greenhouse climate equability has long been a paradox in paleoclimate research. However, recent developments in proxy and modeling methods have suggested that strong seasonality may be a feature of at least some greenhouse periods. Here we present the first multi-proxy record of seasonal temperatures during the Paleogene from paleofloras, paleosol geochemistry, and carbonate clumped isotope thermometry in the Green River Basin (Wyoming, USA). These combined temperature records allow for the reconstruction of past seasonality in the continental interior, which shows that temperatures were warmer in all seasons during the peak early Eocene climatic optimum and that the mean annual range of temperature was high, similar to the modern value (~26°C). Proxy data and downscaled Eocene regional climate model results suggest amplified seasonality during greenhouse events. Increased seasonality reconstructed for the early Eocene is similar in scope to the higher seasonal range predicted by downscaled climate model ensembles for future high-CO$_2$ emissions scenarios. Overall, these data and model comparisons have substantial implications for understanding greenhouse climates



in general, and may be important for predicting future seasonal climate regimes and their impacts
in continental regions.
**1. Introduction**

The Paleogene was the last major greenhouse period in Earth's history and is

characterized by extreme warming events and resultant biological shifts (e.g., Greenwood and
Wing, 1995; Wilf, 2000; Zachos et al., 2001, 2008; McInerney and Wing, 2011), with prolonged
warmth during the early Eocene climatic optimum (EECO) peaking from roughly 52 – 50 Ma
(e.g. Zachos et al., 2008; Hyland et al., 2017). The early Eocene in general is thought to represent
a warm and "equable" global climate state with high mean annual temperatures (MAT; e.g.,
Wilf, 2000; Zachos et al., 2008), low mean annual range of temperatures (MART; e.g., Wolfe,
1978, 1995; Greenwood and Wing, 1995), and low pole-to-equator temperature gradients (LTG;
e.g., Spicer and Parrish, 1990; Greenwood and Wing, 1995; Evans et al., 2018). While high
MAT during the Eocene now seems well established, the feasibility of "equable" conditions
defined by low MART and low LTG is still in question as a result of increasingly complex
global climate models which are unable to reproduce such conditions (e.g., Barron, 1987; Sloan
and Barron, 1990; Sloan, 1994; Huber and Caballero, 2011; Lunt et al., 2012).

Recent proxy work on Paleogene warm intervals and hyperthermals such as the

Paleocene-Eocene thermal maximum (PETM) has suggested that continental interiors may
maintain higher or near-modern MART during these periods, implying that the "low seasonality"
aspect of climate equability may not be reasonable under all greenhouse conditions (e.g., Snell et
al., 2013; Eldrett et al., 2014). Despite this suggestion, it remains unclear whether proxy
estimates from other basins, regions, and greenhouse periods can be reconciled with the range of
feasible conditions provided by climate model studies. Quantitative reconstructions of



seasonality (MART) based on precise proxy estimates of mean annual temperature (MAT),
warm month mean temperature (WMMT), and cold month mean temperature (CMMT) could
help to resolve some of these model-proxy discrepancies by providing a robust and well-
constrained set of seasonal observations for comparison to available climate model outputs.
Robust proxy reconstructions of seasonality are crucial for understanding this aspect of past
greenhouse equability (Lunt et al., 2012; Snell et al., 2013; Peppe, 2013).

Seasonality estimates have previously been made using a variety of proxy

paleothermometers in isolation, and can now be made with higher confidence using recently
developed methods that target each of these individual temperature parameters: MAT can be
estimated using a paleosol geochemistry-based thermometer, WMMT can be estimated using the
carbonate clumped isotope ($\Delta_{47}$) thermometer, and CMMT can be estimated using a nearest
living relative (NLR) floral coexistence thermometer. The bulk major-element geochemistry of
modern soils has been used to quantify the effects of weathering processes via a wide range of
geochemical indices (see Sheldon and Tabor, 2009). The relationship between modern climate
parameters like temperature and indices such as salinization (Sheldon et al., 2002), the paleosol
weathering index (Gallagher and Sheldon, 2013), and the paleosol-paleoclimate model
(Stinchcomb et al., 2016) has led to the development of climofunctions for MAT that have been
used to estimate paleo-MAT during the Cenozoic (e.g., Retallack, 2007; Takeuchi et al., 2007;
Bader et al., 2015; Stinchcomb et al., 2016). The clumped isotope ($\Delta_{47}$) thermometer is based on
the temperature-dependent relative enrichment of multiply substituted isotopologues of $CaCO_3$
($^{13}C^{18}O^{16}O_2$) within the solid carbonate phase, which is independent of the isotopic composition
of the water in which the carbonate precipitated (e.g., Ghosh et al., 2006; Eiler, 2007). For
pedogenic carbonates in temperate regions, this growth temperature is linked to mean warm



season soil temperatures (e.g., Quade et al., 2013; Hough et al., 2014), and has been used to
estimate paleo-WMMT during the Cenozoic (e.g., Snell et al., 2013; Garzione et al., 2014). The
nearest living relative (NLR) coexistence method has been developed based on the sensitive and
highly conserved collective modern cold temperature tolerances of related floras to calculate cold
month temperatures (e.g., Wolfe, 1995; Mosbrugger and Utescher, 1997). Those relationships
have been refined and used to estimate quantitative paleo-CMMT during the Cenozoic (e.g.,
Greenwood et al., 2005; Thompson et al., 2012; Eldrett et al., 2014; Utescher et al., 2014;
Greenwood et al., 2017).

Here we employ a multi-proxy approach using paleosol geochemistry, clumped isotope,

and floral NLR coexistence thermometry methods from the same localities in order to address
seasonality in the past, specifically applying it to the issue of early Eocene greenhouse equability
in the North American continental interior. We estimate MAT, WMMT, and CMMT throughout
the EECO including both defined peak (~51 Ma) and non-peak conditions (e.g., Hyland et al.,
2017), and compare the resultant proxy estimates of temperature seasonality (MART) to the
modern climate state of the region, as well as to downscaled climate model predictions of
temperature seasonality during the Eocene and for future emissions scenarios.

## 2. Methods

The targeted early Eocene locality is the Green River Basin (GRB) in southwestern

Wyoming (USA; Figure 1). The GRB sequence is comprised of a series of terrestrial clastic
rocks deposited during the early Eocene and EECO as a result of Laramide synorogenic fluvial
and lacustrine sedimentation along the margin of endorheic paleo-lake Gosiute (e.g., Clyde et al.,
2001; Smith et al., 2008, 2010, 2015). Contemporaneous multi-proxy records of peak and non-





peak conditions during the EECO are from the interfingering Wasatch Formation, primarily
fluvial sandstones and paleosols of the Ramsey Ranch and Cathedral Bluffs Members, and Green
River Formation, primarily lacustrine shales and carbonates of the Wilkins Peak Member (Figure
1). The paleosols and pedogenic carbonates were sampled from the Honeycomb Buttes near
South Pass, Wyoming (42.24°N, 108.53°W; Hyland and Sheldon, 2013), while the floral
assemblages were sampled from the Latham coal (41.68°N, 107.88°W), Sourdough coal
(41.91°N, 108.00°W), Niland Tongue (41.06°N, 108.77°W), and Little Mountain quarry
(41.28°N, 109.30°W) outside Rock Springs, Wyoming (Figure 1; Wilf, 1998; 2000).

**2.1 Temperature proxies**

**2.1.1 Paleosol geochemistry**

The bulk major-element geochemistry of modern soils (specifically B horizons) has been

used extensively to develop a number of composition-climate relationships, including those
predicted by the paleosol-paleoclimate model ($PPM_{1.0}$), which relates a broad suite of major
element compositions to mean annual temperature (among other factors) at the site of soil
formation (Stinchcomb et al., 2016). Stinchcomb et al. (2016) developed this nonlinear spline
model using the largest available geochemical dataset from 685 modern soils across North
America in order to derive proxy relationships between 11 major and minor oxides and MAT.
This new proxy is calibrated over a wider range of climatic conditions, soil types, and parent
materials than other available proxies (c.f., Sheldon et al., 2002; Gallagher and Sheldon, 2013),
and has been validated via independent comparisons in both modern climosequences
(Stinchcomb et al., 2016) and Miocene paleosols (Driese et al., 2016). Following associated





procedures, our bulk paleosol samples from selected upper Bt horizons of defined Alfisols
(described in detail by Hyland and Sheldon, 2013) were prepared for major-element
geochemistry by cleaning and grinding to a homogenous powder. Samples were analyzed using
lithium borate fusion preparation and X-ray fluorescence (XRF) measurements at ALS Chemex
Laboratory (Vancouver, BC), where analytical uncertainty for analyses was maintained at less
than 0.1% for all elements, and replicate analyses had a mean standard deviation of 0.8% (*Table*
*A.1*). Resultant major and minor elements data were not corrected for loss-on-ignition (e.g.,
Stinchcomb et al., 2016), and were input into the open-access $PPM_{1.0}$ model, which produces
"low", "best" and "high" MAT estimates; we present the "high" estimates as MAT here (see
*Section 4.1* for explanation; *Table A.1*).

**2.1.2 Clumped isotope geochemistry**

The clumped isotope ($\Delta_{47}$) thermometer is based on the theoretical temperature

dependence of the overabundance of multiply substituted carbonate ion isotopologues (primarily
$^{13}C^{18}O^{16}O_2^{-2}$) within the solid carbonate phase, which is independent of the isotopic composition
of the waters from which the carbonate precipitated (e.g., Schauble et al., 2006; Ghosh et al.,
2006; Eiler, 2007). The enrichment of "clumped" isotopologues relative to the abundance
expected for a random distribution of isotopes among isotopologues ($\Delta_{47}$) varies with the growth
temperature of the sampled carbonate (e.g., Ghosh et al., 2006; Dennis et al., 2011; Zaarur et al.,
2013; Kluge et al., 2015; Kelson et al., 2017). Clumped isotope thermometry of soil carbonates is
a useful paleoenvironmental proxy in continental settings (e.g., Eiler, 2011; Quade et al., 2013),
and studies of recent pedogenic carbonates indicate that their clumped isotope values record
environmental temperature conditions during mineral growth. The timing of pedogenic carbonate



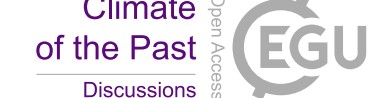

growth is controlled by a combination of soil moisture, $CO_2$, temperature, and other factors over
$10^2$–$10^4$ years (e.g., Cerling, 1984; Cerling and Quade, 1993; Breecker et al., 2009; Zamanian et
al., 2016), and clumped isotope analyses show corresponding variability in recorded
temperatures (e.g., Peters et al., 2013; Hough et al., 2014; Burgener et al., 2016; Ringham et al.,
2016; Gallagher and Sheldon, 2016). However, for pedogenic carbonates forming in forest soils
from mid-latitude regions, this growth temperature has been shown to be linked to mean warm
season soil temperatures in most settings (e.g., Breecker et al., 2009; Passey et al., 2010; Quade
et al., 2013; Garzione et al., 2014; Hough et al., 2014; Ringham et al., 2016), and has been used
to estimate paleo-WMMT during the Cenozoic (e.g., Suarez et al., 2011; Snell et al., 2013;
Quade et al., 2013; Garzione et al., 2014).

Pedogenic carbonate nodules from selected Bk horizons (paleosol depths ~20–240 cm)

were thin-sectioned and analyzed under transmitted light and cathodoluminescence to identify
primary micritic carbonate (Figure 2), which was microdrilled/homogenized for clumped isotope
($\Delta_{47}$) analysis. Extremely shallow (<50 cm) or deep (>200 cm) carbonates were analyzed
specifically to examine temperature depth profiles in paleosols (Figure 2), while pedogenic
carbonates from commonly sampled depths (50–200 cm; e.g., Cerling, 1984; Koch, 1998;
Zamanian et al., 2016) were used for calculating and interpreting paleotemperature records.
Powdered samples and carbonate standards were analyzed in replicate at the University of
Washington's IsoLab, following methods of Burgener et al. (2016) and Kelson et al. (2017),
which are modified after Huntington et al. (2009) and Passey et al. (2010). Briefly, $CO_2$ is
produced from 6–8 mg of pure carbonate reacted in a common phosphoric acid bath (~105%
$H_3PO_4$) at 90°C. Evolved $CO_2$ is then cleaned via passage through a series of automated
cryogenic traps and a cooled (-20°C) Poropak Q column using helium carrier gas through a



nickel and stainless steel vacuum line, and the purified $CO_2$ is transferred to Pyrex break seals.
Each sample is then analyzed on a Thermo MAT253 mass spectrometer equipped with an
automated 10-port tube cracker inlet system and configured to measure m/z 44–49, using data
acquisition methods and scripts presented by Schauer et al. (2016).

All analyses include an automatically-measured pressure baseline (PBL; He et al., 2012),

are corrected using heated gas (1000°C; Huntington et al., 2009) and $CO_2$-water equilibration
(4°C, 60°C) lines during the corresponding analysis period, and are reported in the absolute
reference frame (ARF; Dennis et al., 2011). Following recent work (Daëron et al., 2016; Schauer
et al., 2016), mass spectrometer data are corrected using the $^{17}O$ correction values recommended
by Brand et al. (2010). Carbonate standards for these analyses include international standards
NBS-19 and ETH-2, as well as internal standards C64 and COR, which are all reported relative
to VPDB ($\delta^{13}C$, $\delta^{18}O$) and ARF ($\Delta_{47}$) in *Table B.1*. All samples were analyzed in replicate (3–5)
to minimize standard analytical error, and data were reduced following Schauer et al. (2016).
Carbonate growth temperatures (T[$\Delta_{47}$]) were calculated using the most current and extensive
inorganic calcite calibration (Kelson et al., 2017), which was produced using the updated $^{17}O$
correction values of Brand et al. (2010) and is consistent with our analytical methods. Based on
preliminary comparisons, the Kelson et al. (2017) calibration produces results not significantly
different from data calculated using previous calibrations at moderate Earth-surface temperatures
(Daëron et al., 2016; C. John and M. Daëron, pers. comm., 2016; *Table B.1*).

**2.1.3 Floral coexistence analysis**

Floral physiognomy and floral coexistence techniques are often applied in concert to

arrive at terrestrial paleoclimate estimates (e.g. Spicer et al., 2014; Reichgelt et al., 2015; West et



al., 2015). While floral leaf physiognomy has been used to develop character-climate
relationships for parameters like CMMT and MART (e.g., Wolfe, 1995; Wolfe et al., 1998;
Wing, 1998), other work has raised questions about the reliability of modern calibrations and
possible covariability of seasonal temperatures recorded by floral methods (Jordan, 1997; Peppe
et al., 2010). Similar questions have been raised regarding the nearest living relative (NLR)
coexistence method (Grimm and Denk, 2012; Grimm and Potts, 2016). However, recent
developments have addressed these issues including: 1) improvements or revisions to NLR
assignments for paleofloral assemblages (e.g., Manchester et al., 2014; SIMNHP, 2015), 2) new
global datasets of modern floral distributions (e.g., TROPICOS, 2015; USDA, 2015, GBIF,
2016), 3) high-resolution linked climatic datasets (e.g., Hijmans et al., 2005), and 4) the
application of more rigorous statistical analyses (e.g., Eldrett et al., 2014; Utescher et al., 2014;
Harbert and Nixon, 2015). As a result of this work, bioclimatic analysis has emerged as a refined
version of this approach, employing the climatic range of modern living relatives of plants found
together in a fossil assemblage and statistically constraining the most likely climatic co-
occurrence envelope (e.g., Greenwood et al., 2005; Thompson et al., 2012; Eldrett et al., 2014;
Greenwood et al., 2017).

Fossil assemblages were selected from the literature (e.g., Wilf, 1998, 2000) based on

temporal fit, floristic diversity, and reliable taxonomy. Fossil taxa were each attributed to a
modern taxon based on nearest living relative (e.g., MacGinitie, 1969; Hickey, 1977; Manchester
and Dilcher, 1982; Wolfe and Wehr, 1987; Wing, 1998; Wilf, 1998, 2000; Manchester et al.,
2014; SIMNHP, 2015), with unattributed or disputed placements assigned conservatively at
higher taxonomic levels (*Table C.1*). Climatic envelopes of modern groups in North America
and Asia were retained for the ancient taxa based on environmental niche conservation (e.g.,

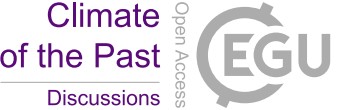



Wang et al., 2010; Fang et al., 2011). Modern taxa distributions (GBIF, 2016) were linked to
high-resolution gridded climatic maps (Hijmans et al., 2005) to extract MAT, WMMT and
CMMT using the Dismo Package in the R Statistical Program (R Core Team, 2013). Prior to
calculating climatic ranges, plant distribution coordinate files were scrutinized for: 1) plants with
dubious taxonomic assignments, as not all identifications were rigorous and not all collected
specimens were taxonomically assigned by experts (only species-level identifications are
included); 2) plants occurring outside of their natural ranges, as many plants occur outside their
adapted environment due to agricultural or aesthetic translocation; and 3) redundant occurrences,
as many duplicate coordinates or researcher entries exist for the same taxon and their inclusion
may skew results toward given localities.

Quantitative paleotemperatures were estimated using a modified bioclimatic analysis

approach (e.g., Greenwood et al., 2005; Thompson et al., 2012; Eldrett et al., 2014; Greenwood
et al., 2017). Overlap ranges of climatic tolerances for coexisting species from each assemblage
were defined by calculating probability density functions of those climatic envelopes (Figure 3
and *Table C.2*) consistent with recent work (e.g., Thompson et al., 2012; Harbert and Nixon,
2015; Grimm and Potts, 2016; Greenwood et al., 2017). In order to avoid inclusion of apparent
coexistence intervals in which no modern occurrence is recorded, we calculate the collective
probability density of taxa co-occurrence for each combination of MAT (x), WMMT (y), and
CMMT (z):
$$f(x|t) = \frac{1}{\sqrt{2\sigma^2\pi}} e^{-\frac{(x-\mu)^2}{2\sigma^2}} \quad (1)$$

$$f(y|t) = \frac{1}{\sqrt{2\sigma^2\pi}} e^{-\frac{(y-\mu)^2}{2\sigma^2}} \quad (2)$$

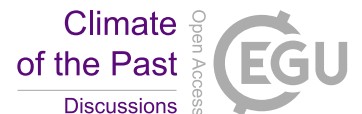

$$f(z|t) = \frac{1}{\sqrt{2\sigma^2\pi}} e^{-\frac{(z-\mu)^2}{2\sigma^2}} \quad (3)$$

$$f(x,y,z,t) = \ln\left[\left(f(x) \times f(y) \times f(z)\right)_{t1} \times ... \times \left(f(x) \times f(y) \times f(z)\right)_{tn}\right] \quad (4)$$

Calculations are repeated such that the likelihood ($f$) is calculated for each climatic combination,
for each taxon ($t$), dependent on the number of taxa ($n$), using the mean and standard deviation of
each taxon (*Table C.2*). Climate input parameters were individual occurrence data points
(~32,000) derived from GBIF (2016), excluding combinations unlikely to represent the climatic
envelope of the taxa in the assemblage by calculating a maximum likelihood probability density
function that defines a precise estimate of temperature parameters with a low standard deviation
for each selected assemblage (Figure 3).

**2.2 Modern climate data and model downscaling**
The modern temperature dataset was derived from 1981–2010 averaged climate normals
from National Oceanic and Atmospheric Administration (NOAA) weather observation stations
within the Green River Basin (n = 18; NCDC, 2010), defined as the area 40.5–43°N by 107–
110.5°W (Figure 1). Future model temperature projection results used a 10-model ensemble
from the Coupled Model Intercomparison Project Phase 5 (CMIP5) under standard low (RCP4.5)
and high (RCP8.5) emissions scenarios (IPCC, 2014). Results were averaged monthly for the
final 10 years of the model run (2090–2099) and calculated over the same study area using
standard bias-correction and spatial downscaling (BCSD) methods developed by PCDMI (2014).
Eocene model temperature results used data from a modified three-dimensional regional climate
model (RegCM3; Sewall and Sloan, 2006; Pal et al., 2007) with established Eocene boundary
conditions including low (560 ppm; LoCO) and high (2240 ppm; HiCO) atmospheric $p\text{CO}_2$
scenarios (Sewall and Sloan, 2006; Thrasher et al., 2009; 2010). Those results were averaged for



the final 20 years of the model run at equilibrium and calculated over the same study area (40.5–
43°N by 107–110.5°W) by integrating data across grid cells monthly for each model year within
the above defined Green River Basin (e.g., Snell et al., 2013). All modern climate normals and
model downscaling results are reported in *Table D.1*.

**3. Results**

PPM$_{1.0}$ statistical model results for MAT from these paleosol samples range from 13.5 to

17.6°C (μ = 15.2°C; σ = 1.3°C). Uncertainty for these estimates is reported as the root mean
squared error of the model fit regression (±2.5°C). Petrographic observation of carbonate nodules
from all depths and selected soils identified dominantly micritic textures with minor components
of sub-angular quartz grains and occasional sparry (>20μm) calcite veins and cements; however,
we were able to identify and micro-sample unaltered fine-grained (<5μm) calcite material in each
of the examined samples (n = 14; Figure 2). Clumped isotope $\Delta_{47}$ values for these samples range
from 0.582 to 0.631‰ (μ = 0.607‰; σ = 0.014‰), which corresponds to an estimated WMMT
range of 18 to 34°C (μ = 25°C; σ = 4°C). Uncertainty for these estimates is reported as
propagated error from analytical and equilibrated $CO_2$ reference frame uncertainty (negligible);
replicate standard error (μ = 0.008‰) or standard error from long-term standards, whichever is
larger; and calibration standard error (e.g., Kelson et al., 2017); which have a combined error
averaging ±3°C. Clumped isotope-based temperature depth profiles in the sampled paleosols
show no clear trend with depth, and estimates are mostly within error for a given paleosol
(Figure 2).  Nearest living relative bioclimatic analysis minimum cold tolerances for these
samples range from -28 to 24°C (μ = 6°C; σ = 7°C), and maximum warm tolerances range from
10 to 43°C (μ = 28°C; σ = 5°C). Probability density functions define bioclimatic envelopes





(Figure 3) corresponding to an estimated CMMT range of 4.2 to 7.6°C ($\mu$ = 5.9°C; $\sigma$ = 1.2°C), an
MAT range of 15.2 to 18.2°C ($\mu$ = 16.5°C; $\sigma$ = 1.1°C), and a WMMT range of 27.9 to 28.7°C ($\mu$
= 28.3°C; $\sigma$ = 0.3°C) for the collective floral assemblages. Uncertainty for these estimates is
reported as $2\sigma$ for individual assemblage PDF distributions, which average ±2°C.  Proxy
estimates from all three methods show a trend of increasing temperatures from non-peak
conditions into the peak EECO (~51 Ma), after which temperatures decreased back to lower
values (Figure 4).

Modern climate normals averaged monthly for the GRB range from -8.4 to 18.1°C, with

a MAT of 4.4°C (*Table D.1*). Downscaled Eocene climate model results averaged monthly for
the GRB range from 4 to 24°C (LoCO) and 6 to 30°C (HiCO), with MATs of 13°C and 16°C,
respectively (*Table D.1*). Downscaled future climate model results averaged monthly for the
GRB range from -5.0 to 20.4°C (RCP4.5) and -2.9 to 24.7°C (RCP8.5), with MATs of 7.1°C and
10.6°C, respectively (*Table D.1*). Monthly temperature trends maintain roughly the same shape
for modern observational data, future model estimates, and Eocene model estimates. However,
the Eocene modeled cases show substantially higher winter temperatures, and in both modern
and Eocene modeled cases the higher emission/$p$CO$_2$ scenario shows an enhanced summer signal
relative to the lower emission/$p$CO$_2$ scenario from the same time period (Figure 5).

**4. Discussion**

**4.1 Temperature estimates**

Temperature estimates from the PPM$_{1.0}$ spline model are based on specifically selected

uppermost B horizons of paleosols with comparable parent materials. These horizons were



selected based on previous work describing and sampling paleosols from the Cathedral Bluffs
Member in the GRB (Figure 1; Hyland and Sheldon, 2013), and based on the characteristics of
soils sampled for the paleosol paleoclimate model dataset (Stinchcomb et al., 2016), in order to
generate the most robust input data for the $PPM_{1.0}$ spline model. While the $PPM_{1.0}$ model
produces multiple possible estimates of paleo-MAT, the estimate shown to be most reliable via
concurrent comparisons with other paleotemperature methods (paleobotanical and paleosol
proxies) is the "high MAT" value we present here (Michel et al., 2014; Stinchcomb et al., 2016;
Driese et al., 2016). We further justify our use of the "high" estimate because the $PPM_{1.0}$ training
dataset heavily samples soils from temperate regions (specifically the conterminous USA) which
tend to have lower MAT (≤10°C) and therefore could place excess weight on low values in the
model predictive space. This sampling bias likely produces the demonstrated pattern of "best"
MAT predictions generally exhibiting positive residuals (Stinchcomb et al., 2016), which means
that the $PPM_{1.0}$ model would be more likely to skew temperature estimates from paleosol and
other modern samples toward lower-than-observed MAT values. The presented mean annual
temperatures appear to coincide with a statistical mean between CMMT and WMMT estimates
(Figure 4), and also agree within uncertainty with independent MAT estimates from other types
of paleosol geochemistry (salinization index, $\delta^{18}O$; Hyland and Sheldon, 2013) and broadly with
updated physiognometric (*Table C.3*; Wilf, 2000) and coexistence analysis paleobotanical
estimates from the GRB (Figure 4).

Based on the assessment of physical and isotopic data, our sampled pedogenic carbonate

nodules appear to be primary records of Earth surface temperatures at the time of their formation.
All sampled nodules preserve micritic carbonate, and transmitted light and cathodoluminescence
images show limited recrystallization or void-filling spar and no evidence of pervasive



remineralization (Figure 2). Isotopic data also suggest primary and uncontaminated carbonate
material; $\Delta_{48}$ values remain low (<<1‰; *Table B.1*), indicating a lack of hydrocarbon or sulfide
contamination (e.g., Guo and Eiler, 2007; Huntington et al. 2009). Temperature and $\delta^{18}O$
measurements remain well within the range of reasonable terrestrial values, particularly for
continental interior basins with seasonal climates (*Table B.1*; e.g., Quade et al., 2013; Hough et
al., 2014). Carbonates forming in temperate regions often exhibit summer/warm-month
temperatures due to warm, dry conditions and low soil $CO_2$ concentrations during those months
(e.g., Breecker et al., 2009; Quade et al., 2013). Such conditions are predicted for the GRB
during the early Eocene based on regional climate models (Thrasher and Sloan, 2009; 2010), and
are evident in paleosol features (Clyde et al., 2001; Hyland and Sheldon, 2013) as well as
evaporative $\delta^{18}O$ of source waters from nearby paleo-lakes Gosiute and Uinta (*Table B.1*; e.g.,
Sarg et al., 2013; Frantz et al., 2014). Further warm biasing of soil temperature with respect to air
temperature can be imparted by radiant ground heating, but such effects are likely negligible in
shaded forest soils (e.g., Quade et al., 2013; Ringham et al., 2016). Clumped isotope data from
two soil depth profiles collected in the GRB agree within uncertainty below ~50 cm (Figure 2),
suggesting that surface heating and depth attenuation of surface temperature variability does not
significantly affect the samples used for our MART reconstructions (paleosol depths ~50–200
cm; e.g., Ringham et al., 2016).

These results imply that the temperatures measured from our pedogenic carbonates

broadly reflect warm month mean soil temperatures (WMMT) as observed in other records (e.g.,
Peters et al., 2013; Hough et al., 2014; Burgener et al., 2016). Possible exceptions are two
samples at the base of the Honeycomb Buttes section (HB-109 and HB-18; *Table B.1*) which
appear to correspond to MAT estimates from the same paleosols (PPM$_{1.0}$; Figure 4). These



lowest temperature estimates from the base of the section may be artificially "cool" as a function
of seasonal precipitation regimes spreading carbonate formation across other parts of the year,
particularly in soils with deeper Bk horizons like these (e.g., Gallagher and Sheldon, 2016).
Because of the likely bias toward MAT in these two samples, we exclude them from calculations
of WMMT or MART as indicated in Figure 4; additionally, this effect means that all of our
clumped isotope-based estimates of WMMT may be artificially low, suggesting that our
calculated MART values could represent a minimum value. However, our resultant clumped
isotope-based temperature estimates are mostly in agreement with both regional climate model
predictions of summer month air temperatures (e.g., Thrasher and Sloan, 2009; Snell et al., 2013)
and paleobotanical coexistence estimates of warm month mean temperatures (Figure 4).
Paleobotanical coexistence methods have been shown to reconstruct paleo-temperatures
robustly, particularly for warm and cold months in well-sampled and taxonomically rich
localities such as these (e.g., Thompson et al., 2012; Grimm and Potts, 2016). However,
uncertainties may be larger than accounted for by the described statistical methods applied to
these assemblages because: 1) many fossil classifications within the GRB assemblages are not
directly comparable to or identifiable as extant species, and coexistence analyses at a generic or
familial level may introduce bias by broadening the temperature tolerance ranges of most groups
(e.g., Wang et al., 2010); and 2) evolutionary or climatic preferences of Paleogene fossil taxa
may not be fully conserved in extant groups, introducing potential sources of error (e.g., Fang et
al., 2011). If we double estimated error to account for these unquantifiable uncertainties, the
collective coexistence probability density functions from these assemblages still produce
CMMT, MAT, and WMMT estimates defined by narrow "maximum likelihood" bioclimatic
envelopes ($< \pm 3°C$; Figure 3; *Table C.2*), which suggest that the environmental characteristics of

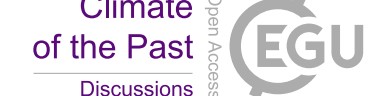

these fossil assemblages are well constrained despite some higher-level NLR assignments.
Additionally, sampling bias from well-sampled temperate regions (e.g., North America) in the
modern GBIF database may place undue weight on the cool end of plant ranges (e.g.,
Greenwood et al., 2017), constraining paleotemperature estimates to lower values or smaller
ranges than is appropriate. This suggests that, similar to clumped isotope-based estimates, our
plant-based MART values could also represent a minimum value. Despite this, paleobotanical
coexistence CMMT estimates agree with regional climate model predictions of winter month
temperatures in the GRB (e.g., Thrasher and Sloan, 2009; 2010), MAT estimates agree broadly
with multiple paleosol-based proxy estimates (Figure 4; Hyland and Sheldon, 2013) and with
updated paleobotanical physiognomy estimates (Figure 4; *Table C.3*; Wilf, 2000), and WMMT
estimates agree with regional climate model estimates (e.g., Thrasher and Sloan, 2009; Snell et
al., 2013) and broadly with clumped isotope-based estimates (Figure 4). Taken together these
proxy results paint a consistent picture of Earth-surface temperatures during the early Eocene,
despite uncertainties inherent in each individual method.

**4.2 Temperature seasonality**

Because each of these proxies appears to represent different seasonal temperatures

robustly, we combine these estimates to produce a new multiply constrained investigation of
paleo-MART. By calculating the differences between CMMT from paleobotanical coexistence
analysis, MAT from paleosol geochemistry or paleobotanical analyses, and WMMT from $\Delta_{47}$
composition or paleobotanical coexistence analysis, we can directly estimate MART in the past
and compare differences in seasonal temperatures independent of calculation method (c.f., Snell
et al., 2013). In other words, our approach can define MART as: 1) the difference between



WMMT and CMMT; or 2) twice the difference between MAT and either WMMT or CMMT,
assuming that MAT falls half way between those estimates by definition (Table 1). Because our
approach can calculate MART using both methods and an average of multiple proxies, this
allows for a wide range of independent checks on our estimates, providing the most robust
available paleo-MART (Table 1). Each method provides consistent answers that are statistically
indistinguishable for a given time period (Student's $t$-test p-values = 0.4–0.9), lending
confidence to calculations which show that MART ranged from 21–26°C during the early
Eocene (Table 1). MART was generally slightly lower than modern (~21–23°C) across this
interval, but appears to have increased to near-modern ranges during the peak EECO (~26°C;
Figure 4; Table 1). The calculated uncertainty of the difference between these populations
(S.E.D) is ~4°C, which makes the non-peak and peak intervals statistically distinct though nearly
overlapping.

Estimates from the lower end of our reconstructed MART range are still higher than

MART estimates from individual paleobotanical proxies (15–18°C; e.g., Greenwood and Wing,
1995; Wolfe et al., 1998), but compare favorably to estimates from regional climate models with
assumed lacustrine or paludal land cover (20–22°C; Thrasher and Sloan, 2010). However,
estimates from the higher end of the reconstructed MART range compare more favorably to
modeled MART values with assumed woodland or forested land cover (24–26°C; Thrasher and
Sloan, 2010; Snell et al., 2013). The transient nature of paleo-lake Gosiute and the variable
evolution of environments within the GRB throughout the early Eocene is well documented in
stratigraphic archives, indicating that the basin may have been alternately dominated by the
paleo-lake or by forested floodplains during this period (Smith et al., 2008; 2014). In this
context, our results suggest that both lower (though still in excess of any previous paleobotanical





estimates) and higher MART states may in fact be reasonable for this region at different points
during the early Eocene as the GRB evolved. Moreover, proxy and modeling work does not
appear to be contradictory, instead having captured different portions of the range of possible
MART values indicated for the peak vs. non-peak EECO in this part of the continental interior
(Figures 4 and 5). Regardless, these results suggest that MART values lower than ~20°C (e.g.,
Greenwood and Wing, 1995; Wolfe et al., 1998) may be unreasonable during any part of the
EECO, even in the context of variable climate and environmental conditions.

**4.3 Seasonality implications**

Our new proxy data and model comparisons have important implications for continental

climates, as they suggest two potential characteristics of seasonality in interior regions during
warming events: 1) proxies tend to indicate continental temperatures on the high end of modeled
ranges in all seasons, and 2) both proxies and regional models indicate that summer temperatures
may increase disproportionately, actually broadening MART, at high atmospheric $p$CO$_2$. While
proxy and model estimates of paleotemperature generally agree through the early Eocene in the
GRB, proxy estimates consistently fall in the top half of all modeled values (Figure 5). Although
these model and proxy results are not statistically distinct, they may suggest that realistic
environmental responses could have a skewed distribution within the range of model-predicted
climate outcomes, an observation which has been made previously for other regions and time
periods (e.g., Roe and Baker, 2007; Diffenbaugh and Field, 2013).

Winter temperatures were generally high during the Eocene (Figures 4 and 5; e.g.,

Greenwood and Wing, 1995), but during the peak EECO summer temperatures appear to have
increased disproportionally, broadening the range of MART (Figures 4 and 5). Regional Eocene



climate model output for the GRB predicts lower MART (~20°C) under low $pCO_2$ conditions

(LoCO scenario), and higher MART (~24°C) under high $pCO_2$ conditions (HiCO scenario;

Figure 5; *Table D.1*). Therefore, a theoretical transition from lower (≤500 ppm) to higher (≥1000

ppm) atmospheric $pCO_2$ during the peak EECO (e.g., Hyland and Sheldon, 2013; Jagniecki et al.,

2015) could effectively broaden MART and result in extreme summer temperatures during that

period, which would be consistent with both regional model and proxy predictions in the GRB

(Figure 5).  Regional model-proxy agreement on the plausibility of variable moderate to high

MART (20–26°C) in continental interiors fits with global simulations employing a reasonable set

of radiative forcings and climate sensitivities, which project similar seasonality ranges during

this and other greenhouse events (Huber and Caballero, 2011; Lunt et al., 2012). These

temperature seasonality estimates also corroborate recent work on other regions and warm

periods (e.g., Snell et al., 2013; Eldrett et al., 2014), and further support the interpretation that

continental interiors were less "equable" than previously thought under greenhouse conditions

(Snell et al., 2013; Peppe, 2013).

Increased seasonality and the disproportionate response of summer temperatures during

greenhouse climates also has significant implications for predicting future change in continental

interiors. Current projections for the next century using downscaled global climate model

ensembles (PCDMI, 2014; *Table D.1*) indicate generally increased temperatures and changing

seasonality in North America, and GRB temperatures are projected to increase particularly

during winter months (Figure 5). However for high emissions scenarios that may be closer in

character to greenhouse conditions like the peak EECO or the PETM (RCP8.5; e.g., IPCC, 2007;

Lunt et al., 2012), summer temperatures in the GRB increase more strongly, broadening MART

(Figure 5; *Table D.1*). This trend in MART from peak EECO proxy data and high-





emission/$p$CO$_2$ model simulations in both the future and Eocene suggests a potential atmospheric
$p$CO$_2$ threshold for enhanced seasonality, and provides support for models and observations
indicating that continental interiors may experience more extreme seasonality in the future under
heightened greenhouse conditions (e.g., IPCC, 2007; Diffenbaugh and Field, 2013; Diffenbaugh
et al., 2017). The mechanism for producing this increased seasonality remains unclear and
requires further study in terms of both proxy applications and model development, although
changes in land cover may play a crucial role at least in regional variability (Thrasher and Sloan,
2010; Diffenbaugh and Field, 2013).

**5. Conclusions**

Estimates of winter (paleofloral NLR coexistence), mean (paleosol geochemistry), and

summer (clumped isotope) temperatures from the early Eocene in the Green River Basin of
Wyoming (USA) provide new multi-proxy constraints on seasonality (mean annual range of
temperature) in terrestrial settings during greenhouse periods. These records show that MART
was variable but near (or above) modern values during the early Eocene climatic optimum,
confirming both that seasonality in continental interiors may not remain constant, and that EECO
conditions likely do not conform to at least the seasonality aspect of greenhouse "equability".
Comparisons between proxy data and regional/downscaled climate models further imply that
temperature seasonality may respond differently at low vs. high atmospheric $p$CO$_2$. Overall, this
suggests that our understanding of past greenhouse climates in continental interiors may be
incomplete when it comes to "equability", and proposes the potential for extreme seasonality in
these regions during past warming events and in the future, which likely has important
implications for natural ecosystems and human infrastructure.



**Data Availability.** Summarized paleosol, isotope, floral, and modeling data are available in the
Supplement, and detailed sample or locality data are available from the authors on request.

**Author Contributions.** EGH, NDS, and KWH designed the study; EGH and NDS collected the
samples and conducted fieldwork; EGH conducted laboratory analyses; EGH, KWH, and TR
conducted data analyses and reduction; all authors contributed to the writing of the manuscript.

**Competing Interests.** The authors declare that they have no conflicts of interest.

**Acknowledgments.** The authors thank D. Peppe, K. Snell, and XXX for manuscript comments;
A. Schauer, L. Burgener, and P. Wilf for assistance with proxy data; PCDMI, WCRP, L. Sloan,
and J. Sewall for archived modeling datasets; NSF grants EAR-1252064 and 1156134 (KWH),
GSA's Farouk El-Baz grant (EGH), and the Quaternary Research Center and Future of Ice
Initiative at the University of Washington for funding and support.



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



**Figure Captions.**
**Figure 1.** Map and stratigraphy of the Green River Basin. A) Map of the region, showing major
sedimentary basins and topographic highs. Stars show proxy record sampling sites (paleosols in
yellow, paleoflora in red), and dashed box is the sampling region for modern climate stations and
the downscaling domain for both models. CF = Cordilleran fold-thrust belt, UU = Uinta uplift,
WR = Wind River uplift, OC = Owl Creek uplift, GM = Granite Mountains, FR = Front Range.
B) Simplified stratigraphy of the central to eastern GRB, showing facies for the Green River
Formation (GRF) and the equivalent and interfingering Wasatch Formation (WF) based on the
work of Smith et al. (2015) and Hyland and Sheldon (2013). LY = Lysitean, BF = Blacksforkian,
LU = Luman Member, NT = Niland Tongue, TM = Tipton Member, WPM = Wilkins Peak
Member, LA = Laney Member, RR = Ramsey Ranch Member, CB = Cathedral Bluffs Member.

**Figure 2.** Paleosol carbonate descriptions. A) Paired transmitted light and cathodoluminescence
(CL) images of carbonate nodules showing primary micrite in sampled nodules (I-II) and
diagenetically altered material in unsampled nodules (III-IV). Images taken on a Premier ELM-
3R Luminoscope at 8–10 kV, 0.5 mA, and 6.6–13.3 Pa with preset 1 s exposure; scale bars ~50
μm. B) Clumped isotope-based soil temperature profiles from discrete layers sampled within
analyzed paleosol exemplars. Profile HB-129 contained nodular carbonate layers at 20–30 cm,
50–65 cm, and 80–100 cm; Profile HB-187 contained nodular carbonate layers at 150–170 cm,
190–205 cm, and 240–260 cm.

**Figure 3.** Floral methods description. A) Probability density functions of hypothetical Taxa A
and B along climatic variable X to form a PDF representative of the maximum likelihood of co-



occurrence. B) Hypothetical climatic envelope of Taxon Q with climatic variables X and Y,
where point R occurs outside the envelope of Taxon Q but within its range of both variables
(creating a false inclusion of point R). C) Probability density function distributions for seasonal
temperatures from sampled paleofloral sites, where arrows indicate calculated mean
temperatures for each parameter.

**Figure 4.** Temperature proxy estimates of CMMT (white), MAT (gray), and WMMT (black)
through the early Eocene. Triangles represent paleobotanical coexistence estimates, squares
represent paleosol geochemistry estimates, stars represent revised paleobotanical physiognomy
estimates, and circles represent clumped isotope estimates. Error bars represent PDF 2σ
(paleobotanical coexistence), root mean squared error ($PPM_{1.0}$ paleosol geochemistry),
calibration standard error (paleobotanical physiognomy), and propagated analytical/calibration
error (clumped isotopes). Shading highlights peak EECO conditions based on previous work
(e.g., Hyland et al., 2017), and dashed line highlights exclusion of two data points (see
Discussion). Estimates of peak EECO (51 ±0.5 Ma) and non-peak EECO MART are defined as
described in Table 1 and the Discussion, with MAT shown by vertical lines. Modern MART and
MAT are from averaged climate normals for NOAA weather stations in the GRB (NCDC, 2010).

**Figure 5.** Averaged monthly mean temperatures in the GRB, including: modern instrumental
data (filled black circles; NCDC, 2010); high (red squares; RCP8.5) and low (red circles;
RCP4.5) future emissions scenarios (PCDMI, 2014); high (blue squares; HiCO) and low (blue
circles; LoCO) early Eocene $pCO_2$ scenarios (Thrasher and Sloan, 2009; 2010); and proxy
reconstructions of WMMT and CMMT for non-peak (filled triangles) and peak EECO (open



triangles) from this study. Method-averaged MART estimates shown for each category
(symbols/colors match main panel).

**Table 1.** Comparison of Eocene MART estimates using different constraining temperatures and
calculation methods.


**Supplement.**
**A. Paleosol Data**
*Table A.1.* Paleosol geochemistry data
**B. Isotope Data**
*Table B.1.* Clumped isotope data summary
*Table B.2* Clumped isotope data full
**C. Floral Data**
*Table C.1.* Floral lists and NLR data
*Table C.2.* Climatic envelopes for plant taxa
*Table C.3.* Mean annual temperature estimates
**D. Modeling Data**
*Table D.1.* Modern climate data and model outputs



Figure 1.

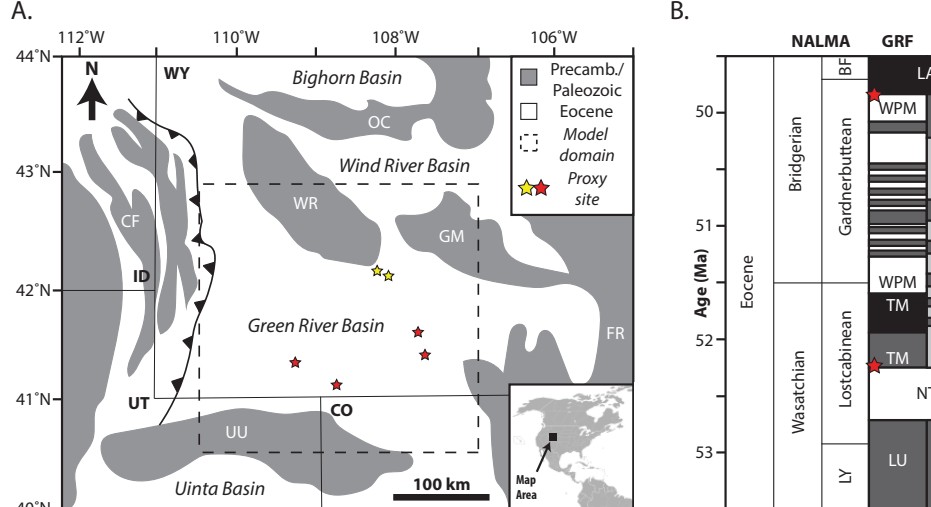

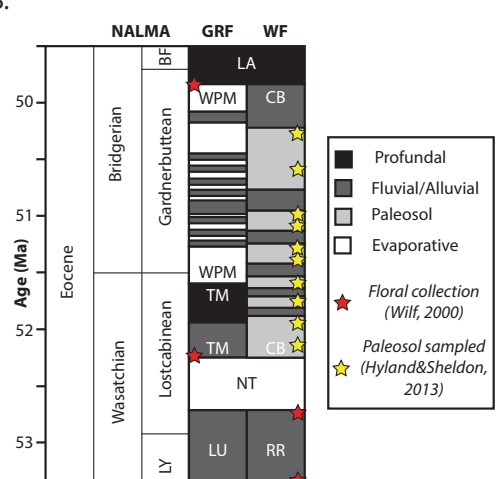





Figure 2.

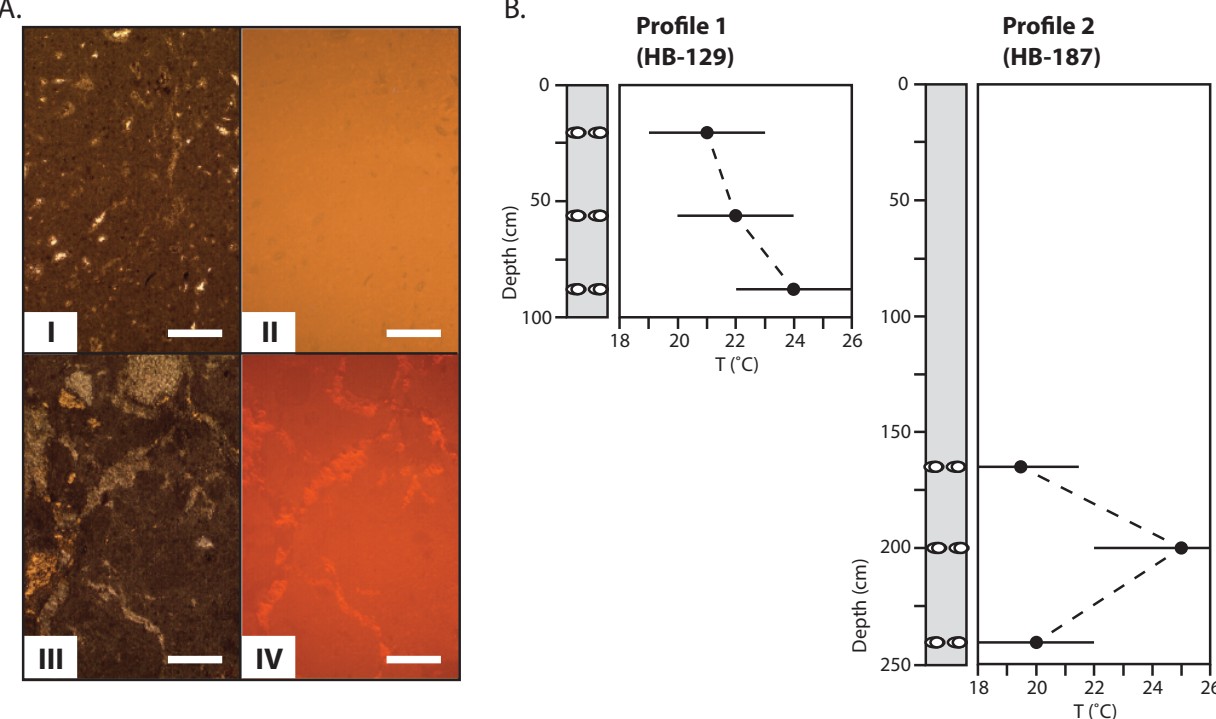



Figure 3.





Figure 4.

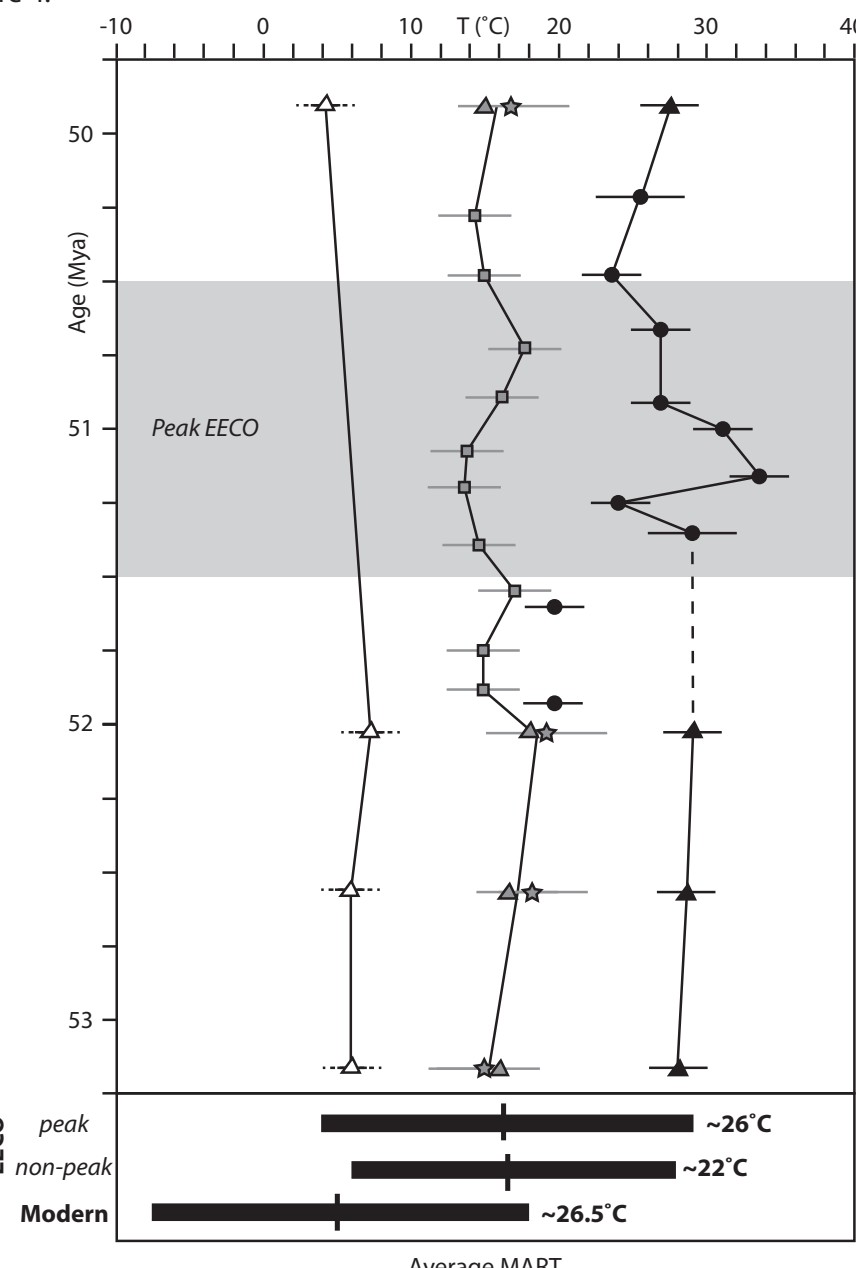




Figure 5.

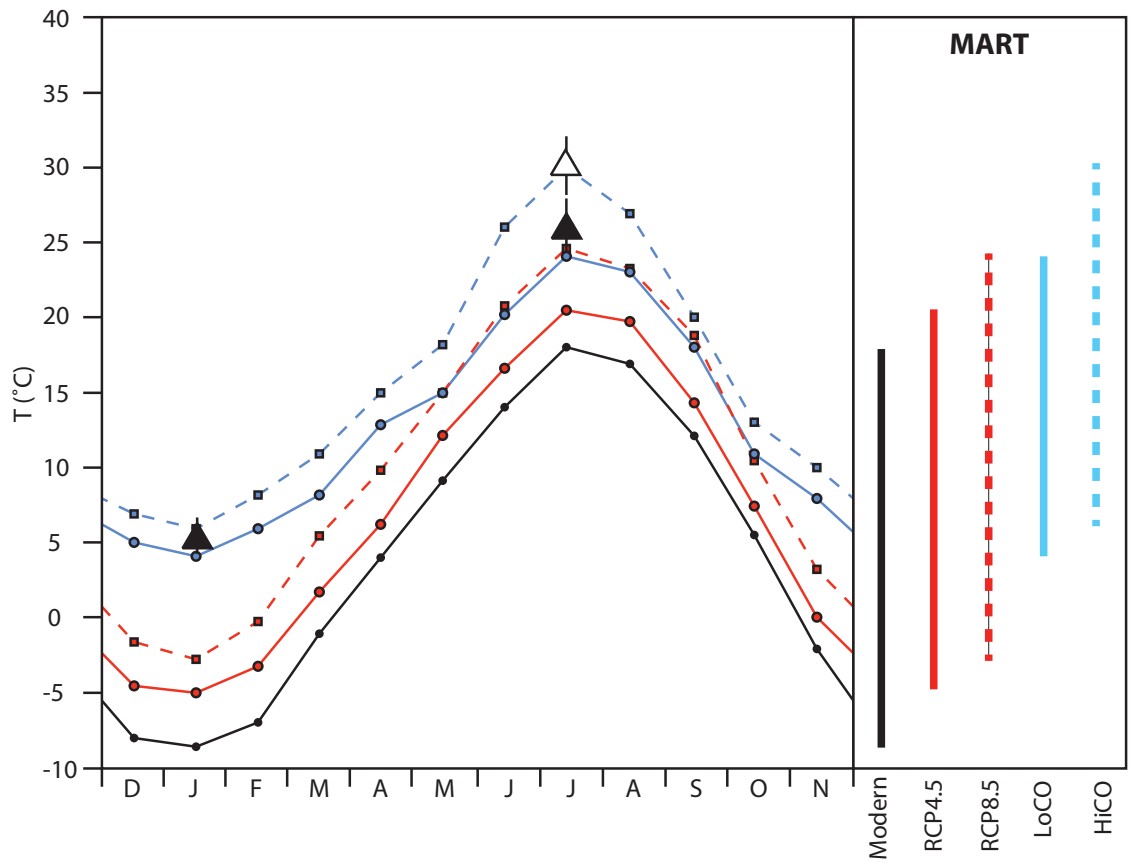

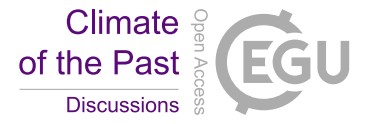

Table 1.

| Interval | CMMT* | MAT* | WMMT* | MARTT† | MARTC† | MARTW† |
|---|---|---|---|---|---|---|
| **peak EECO** <br> *(50.5 - 51.5 Ma)* | -- | 15.4°C | 28.2°C | -- | -- | 26°C (4) |
| **non-peak EECO** <br> *(53.5 - 51.5 Ma &* <br> *50.5 - 49.5 Ma)* | 5.9°C | 15.6°C | 26.8°C | 22°C (1) | 21°C (1) | 23°C (1) |

MARTT = WMMT-CMMT
MARTC = (MAT-CMMT)×2
MARTW = (WMMT-MAT)×2
\* Average of all available temperature proxy data across indicated time interval.
† Average MART estimate for each calculation method, number in parentheses is S.D. of calculation group.