# Peer review of "interior during the early Eocene climatic optimum"

_Climate of the Past, 2018_

## Referee Comment (RC1) · G. Retallack (Referee) · 10 Apr 2018

This is an excellent paper comparing seasonality changes in paleoclimate through the Paleocene-Eocene thermal maximum, and finding that the greenhouse spike did not have an equable climate compared with before and afterward, as some have predicted. This is an important qualification for understanding climatic change in a higher CO2 world, in emphasizing seasonality of temperatures rather than averages. This paper is excellent and publishable with minor revision. I am familiar with most of the methods deployed and consider them skillfully applied to the problem. The authors have a high level of technical competence and understanding of limitations of each method. My

main reservation is that I do not agree that the difference (ca 4oC) in temperature seasonailty of the greenhouse spike compared with the times before and afterward is significant, given standard errors on the various proxies. I consider the temperature seasonality before, during, and afterward statistically indistinguishable. This is not quite the same as the interpretation given, but does make their point that greenhouse spike temperatures were far from equable. A minor quibble, is that my understanding of nearest living neighbor and other paleobotanical estimates of paleoclimates rely on an adequate number of species in the assemblage (usually at least 30 species). The number of taxa in each assemblage should be reported, perhaps most conveniently in the figure.

———————————————

---

## Author Comment (AC1) · 7 Jun 2018

1) This is an excellent paper comparing seasonality changes in paleoclimate through the Paleocene-Eocene thermal maximum, and finding that the greenhouse spike did not have an equable climate compared with before and afterward, as some have predicted. This is an important qualification for understanding climatic change in a higher CO2 world, in emphasizing seasonality of temperatures rather than averages. This paper is excellent and publishable with minor revision. I am familiar with most of the methods deployed and consider them skillfully applied to the problem. The authors have a high level of technical competence and understanding of limitations of each

method.

We would like to thank Dr. Retallack for his time in reviewing our manuscript, and for the positive evaluation of our work and its importance within the context of research into greenhouse periods.

2) My main reservation is that I do not agree that the difference (ca 4°C) in temperature seasonality of the greenhouse spike compared with the times before and afterward is significant, given standard errors on the various proxies. I consider the temperature seasonality before, during, and afterward statistically indistinguishable. This is not quite the same as the interpretation given, but does make their point that greenhouse spike temperatures were far from equable.

We too were concerned about the significance of the trend across the peak EECO due to the larger error inherent in these measurements, however we believe we have addressed this concern via the combination of 1) averaging larger numbers of replicates (e.g., Fernandez et al., 2017), and 2) applying common statistical tests (e.g., uncertainty of difference). By binning our temperature data into "peak" and "non-peak" periods we allow for larger datasets in calculating a mean value, which leads to lower error in the estimation of those means (Fernandez et al., 2017). Thus, our mean estimates for the two periods has a lower error than any individual estimate from the record. Additionally, we applied an uncertainty of difference test to these two populations, and find that the difference between the two populations is greater than the uncertainty, suggesting that the two populations are statistically distinguishable and not a result of high uncertainty (as described in Section 4.2). We do also agree with Retallack that regardless of whether the peak EECO has a higher MART than the non-peak period, both periods suggest that greenhouse conditions do not lead to "equability" of seasonality, which is our primary conclusion.

3) A minor quibble, is that my understanding of nearest living neighbor and other paleobotanical estimates of paleoclimates rely on an adequate number of species in the

assemblage (usually at least 30 species). The number of taxa in each assemblage should be reported, perhaps most conveniently in the figure.

We agree with Dr. Retallack that the number of taxa is an important consideration for examining the climatic tolerances of assemblages, and will add this information to Figure 3 before final publication of the manuscript. The number (and type) of taxa for each assemblage was previously reported in the supplemental information (Table C1), but should also have been included in the manuscript itself.

---

## Referee Comment (RC2) · N. Barbolini (Referee) · 12 Jul 2018

This manuscript is a significant contribution to the field, combining multiple temperature proxies in an interdisciplinary fashion to reconstruct climatic ranges over the EECO. Bringing the focus onto detecting seasonality in the fossil record provides a more nuanced representation of past climates. Palaeobotanical temperature reconstructions are performed using the latest, statistically rigorous methodology, and an important synthesis on state-of-the-art is included. The presentation and quality of the work are excellent. The figures are well-illustrated and comprehensive.

It is a pity that no CMMT (floral) data are available for the peak EECO. In order to

represent this more realistically, I suggest modifying the straight line joining the two CMMT data points, separated by more than 2 Mya, in Figure 4. It could be represented either by a dotted line explained in the figure caption as an aliased signal, or omitted completely. Other relevant points have already been raised by the other reviewer and resolved by the authors. The manuscript is publishable subject to this and the following technical corrections:

1. Line 89: "The GRB sequence is comprised of a series of terrestrial clastic..."

   - Correct to either "sequence is composed of" or "sequence comprises"

2. Supplement, Table C.1:

   - The heading "Floral group" should change to "Taxa"
   - When taxa are only identifiable to genus level, "sp." should be reformatted without italics e.g. *Sabalites* sp.
   - Taxa could be ordered alphabetically for easier future comparisons of floral lists

---

## Author Comment (AC2) · 16 Jul 2018

1) This manuscript is a significant contribution to the field, combining multiple temperature proxies in an interdisciplinary fashion to reconstruct climatic ranges over the EECO. Bringing the focus onto detecting seasonality in the fossil record provides a more nuanced representation of past climates. Palaeobotanical temperature reconstructions are performed using the latest, statistically rigorous methodology, and an important synthesis on state-of-the-art is included. The presentation and quality of the work are excellent. The figures are well illustrated and comprehensive.

We would like to thank Dr. Barbolini for her time in reviewing our manuscript, and for

the positive evaluation of our work and its importance within the context of paleoclimate reconstructions.

2) It is a pity that no CMMT (floral) data are available for the peak EECO. In order to represent this more realistically, I suggest modifying the straight line joining the two CMMT data points, separated by more than 2 Mya, in Figure 4. It could be represented either by a dotted line explained in the figure caption as an aliased signal, or omitted completely.

We agree that it is unfortunate that there are no known preserved floral assemblages in this region during the peak EECO. As suggested, we will modify Figure 4 before final publication to include a dashed line between the youngest two assemblages, and will update the figure caption to explain the possibility of an aliased signal during this extended period.

3) Other relevant points have already been raised by the other reviewer and resolved by the authors. The manuscript is publishable subject to this and the following technical corrections:

Line 89: "The GRB sequence is comprised of a series of terrestrial clastic...": Correct to either "sequence is composed of" or "sequence comprises".

We have updated the manuscript text to read "sequence is composed of" as per the reviewer.

Supplemental Table C.1: a) The heading "Floral group" should change to "Taxa"; b) when taxa are only identifiable to genus level, "sp." should be reformatted without italics (e.g. Sabalites sp.); c) taxa could be ordered alphabetically for easier future comparisons of floral lists.

We agree with points A and B of the suggested changes to this supplemental table, and will update Table C.1 before final publication. With regard to point C, we list taxa in Table C.1 in this order to maintain the format of the originally published floral lists from these

sites (Wilf, 2000), as we believe this will minimize confusion in future comparisons which may include both works. This original format lists taxa in order of abundance at each locality, an observation that we will add to the descriptive text of the table before final publication.

---

## Referee Comment (RC3) · Anonymous Referee #3 · 1 Aug 2018

Review of manuscript cp-2018-28

The manuscript 'Temperature seasonality in the North American continental interior during the early Eocene climatic optimum' by Hyland *et al*. presents new clumped isotope, soil geochemistry and paleofloral data from a section spanning the EECO from the Green River Basin. The authors use these proxies to reconstruct both mean annual, winter, and summer temperatures, and therefore to reconstruct Eocene seasonality in this region, and how seasonality changed over the EECO. Perhaps the most interesting finding is that seasonality during the peak EECO was similar to today, calling into question the hypothesis that past greenhouse intervals such as this were characterised by an 'equable' climate (warm/hot, but low seasonality). The data are interesting, and the subject of Eocene climate is certainly of relevance to a wide number of researchers across several disciplines. Therefore, the manuscript would certainly be suitable for *Climate of the Past*. Nonetheless, I offer several suggestions for clarification, especially regarding uncertainty in the seasonality reconstructions. Lastly, I suggest that the model and discussion surrounding future climate change detracts from the focus of the manuscript and could be omitted.

**Substantive comments**
- Overall the authors do an excellent job of considering the uncertainties in the proxies utilised here, and as such I recommend that there should be clarification in the discussion of what is being reconstructed. My main issue is that there are no data on cold month mean temperature (CMMT) for the EECO, such that seasonality, i.e. the difference between WMMT and CMMT, is assumed to be twice the difference between mean annual temperature (MAT) and WMMT. The authors are honest about this (Tab. 1), and I agree that it is encouraging that MAT falls approximately midway between CMMT and WMMT in the pre-peak-EECO interval, but there is no *a priori* reason to assume that CMMT did not increase disproportionately during the EECO, and as such EECO seasonality is not really constrained here. I recommend rewording the abstract, discussion and conclusions accordingly.
- I found the discussion of seasonality under future climate scenarios to be a distraction from the rest of the manuscript. My suggestion would be to deal with this topic separately given that the Eocene data do not necessarily inform us of the accuracy of the CMIP5 model ensemble into the future. A more interesting comparison, if the data are available, would be to compare these seasonality reconstructions to the EoMIP model ensemble rather than RegCM3 alone (which would also allow some uncertainty to be placed on the model seasonality).

**Minor comments**
- Section 2.1.1. For people like me, who are not familiar with the details of the use of soil geochemistry as a paleoclimate indicator, it would be useful to include a few sentences on how robust this proxy is to diagenetic alteration (e.g. through interaction with groundwater).
- Line 167. Please specify what you mean by 'are corrected' in this sentence. Do you mean that following the PBL correction the $\delta_{47}$-$\Delta_{47}$ slope is also used to make a correction? If so, what is the gradient?
- Lines 177-180. Please clarify these sentences. The Kelson equation would certainly result in significantly different temperatures compared to at least some other calibrations [e.g. Zaarur *et al*., 2012], depending on what is meant by 'moderate'. And why preliminary?
- Section 2.2. The introduction and description of the model results is extremely brief which is ok given that these data are previously published (although a few more details would be helpful), but what I did miss was an explanation of why this model was chosen for comparison. Why not one (or the ensemble) of EoMIP simulations?

- Line 317. It would be useful to elaborate on what is meant by 'isotopic data'. Is it that the clumped isotope results look reasonable, or do you mean calculated $\delta^{18}O_w$ look reasonable? If the latter, state what these are in the text in this paragraph (perhaps around lines 323-326).
- Lines 363-366. This sounds encouraging except that a $\pm3°C$ uncertainty in both WMMT and CMMT results in a seasonality uncertainty of $6°C$, given that presumably there may be systematic biases across the reconstructed interval?
- Section 4.2. Somewhere near the beginning of this section please state how EECO and pre-EECO seasonality were calculated. Are the seasonality estimates derived by comparing the mean of (e.g.) all CMMT data to the mean of all WMMT data for that interval? Judging by eye, there seems to be no difference between EECO and pre-peak-EECO MAT and WMMT if all the data are averaged.
- Lines 397-398. It was not clear to me the first time I read this whether the numbers in brackets refer to the modern or Eocene seasonal range, especially on line 397, reword for clarity.
- Lines 396-401. As you state earlier, both MAT and WMMT may be cool-biased though, so the agreements and uncertainties discussed here only apply if this is not the case. Consider reemphasising this.
- Lines 433-435. This sentence is worded too strongly. There are no reconstructions of CMMT during the EECO, but moreover the WMMT proxy in the EECO is different to the WMMT proxy in the pre-peak-EECO interval. This latter issue is especially problematic given that the clumped and paleobotanical evidence are not in agreement in the interval for which both are available. I fully sympathise with the authors in that continental climates are difficult to reconstruct, and as I said before, the recognition and discussion of uncertainties in this manuscript is excellent overall. However, I recommend stating these issues more clearly at this point in the text including a more thorough discussion of whether the peak and pre-peak EECO seasonalities are really distinguishable from each other.
- Figure 4. Either remove the left half of the peak EECO bar at the bottom of the figure or shade it white so that it is clear that this has not been reconstructed. Please clarify whether the tick marks represent the midpoint between CMMT and WMMT or whether they are placed at the location of the mean MAT estimates.
- Supplement. Please given ages with all reconstructions and not just heights, or at the very least state at which heights the peak EECO occurred.

**Typos**
- Line 122 'elements'.

---

## Author Comment (AC3) · 24 Aug 2018

The manuscript 'Temperature seasonality in the North American continental interior during the early Eocene climatic optimum' by Hyland et al. presents new clumped isotope, soil geochemistry and paleofloral data from a section spanning the EECO from the Green River Basin. The authors use these proxies to reconstruct both mean annual, winter, and summer temperatures, and therefore to reconstruct Eocene seasonality in this region, and how seasonality changed over the EECO. Perhaps the most interesting finding is that seasonality during the peak EECO was similar to today, calling into question the hypothesis that past greenhouse intervals such as this were characterised by an 'equable' climate (warm/hot, but low seasonality). The data are interesting, and the subject of Eocene climate is certainly of relevance to a wide number of researchers across several disciplines. Therefore, the manuscript would certainly be suitable for Climate of the Past.

We would like to thank the reviewer for their time in reviewing our manuscript, and for the positive evaluation of our work and its relevance for multiple disciplines.

Substantive Comments Overall the authors do an excellent job of considering the uncertainties in the proxies utilised here, and as such I recommend that there should be clarification in the discussion of what is being reconstructed. My main issue is that there are no data on cold month mean temperature (CMMT) for the EECO, such that seasonality, i.e. the difference between WMMT and CMMT, is assumed to be twice the difference between mean annual temperature (MAT) and WMMT. The authors are honest about this (Tab. 1), and I agree that it is encouraging that MAT falls approximately midway between CMMT and WMMT in the pre-peak-EECO interval, but there is no a priori reason to assume that CMMT did not increase disproportionately during the EECO, and as such EECO seasonality is not really constrained here. I recommend rewording the abstract, discussion and conclusions accordingly.

We fully understand and discuss the fact that we do not have CMMT estimates for this whole period (as shown in Figure 4 and Table 1); however, we disagree that this means "EECO seasonality is not really constrained here". We argue that the applied method of calculating MART as twice the difference between WMMT and MAT is robust for multiple reasons: 1) this calculation is common practice for estimating seasonality, and has been shown to be robust in comparisons to both paleobotanical and model-derived estimates (Snell et al., 2013; Suarez et al., 2017; Kelson et al., 2018); 2) as described in this work (Section 4.2), for periods where CMMT results do exist, the two methods (WMMT-CMMT and 2*[WMMT-MAT]) produce statistically indistinguishable results; and 3) calculations from modern environments worldwide using ERA-Interim climate reanalysis outputs consistently show that for all vegetated land surfaces, the

difference between MART calculations derived from these two methods is «1°C (Burgener et al., in review). As a result, we believe that seasonality is indeed constrained by our chosen method for both the peak and non-peak EECO in this work, and therefore do not rewrite the Abstract/Discussion/Conclusions (except to improve clarity on this point).

I found the discussion of seasonality under future climate scenarios to be a distraction from the rest of the manuscript. My suggestion would be to deal with this topic separately given that the Eocene data do not necessarily inform us of the accuracy of the CMIP5 model ensemble into the future. A more interesting comparison, if the data are available, would be to compare these seasonality reconstructions to the EoMIP model ensemble rather than RegCM3 alone (which would also allow some uncertainty to be placed on the model seasonality).

We believe that the comparison to seasonality under future climate scenarios is a key implication for this work, and that the inclusion of this comparison is important for piquing interest in the continued study of seasonality as part of understanding other climate regimes. We agree that it should not distract from the Eocene work, which is why we have limited discussion of that comparison to its own section at the end of the manuscript (Section 4.3). We do not intend for the Eocene data to inform about the accuracy of CMIP5 models (indeed we do not think that appropriate), merely to show that both sources suggest similar patterns of increased seasonality under greenhouse conditions. While we agree that comparing our Eocene data or reported CMIP5 results to an Eocene ensemble such as EoMIP would be ideal, the output for such a comparison does not currently exist. The reason we chose to use the RegCM3 regional model is that it provides high spatial resolution over this region (50 x 50 km), allowing us to make an accurate comparison of GRB proxy data with a model of the GRB itself. While the models of the EoMIP ensemble perform well on a global scale, their grid cell size is far too large (3.75 x 2.5°) to capture the characteristics of this continental region, and thus would be inappropriate for this work. Future work examining seasonality on a broader

scale using data from multiple localities could certainly benefit from comparisons to the EoMIP ensemble, however we believe this is beyond the scope of our current study.

Minor Comments Section 2.1.1. For people like me, who are not familiar with the details of the use of soil geochemistry as a paleoclimate indicator, it would be useful to include a few sentences on how robust this proxy is to diagenetic alteration (e.g. through interaction with groundwater).

As suggested by the reviewer, we have added text to Section 2.1.1 describing the robustness of soil geochemical proxies, which have been widely tested in a large number of environments and on timescales spanning the Cenozoic and beyond (Sheldon and Tabor, 2009 and references therein).

Line 167. Please specify what you mean by 'are corrected' in this sentence. Do you mean that following the PBL correction the $\delta47$-$\Delta47$ slope is also used to make a correction? If so, what is the gradient?

Yes, we mean that we follow the common practice of using the slope of heated and water equilibrated $CO_2$ to normalize values (Huntington et al., 2009) and place them into the ARF (Dennis et al., 2011). The slopes of these lines are reported in the supplement (Tables B.1 and B.2), and we have added a reference to this in the text.

Lines 177-180. Please clarify these sentences. The Kelson equation would certainly result in significantly different temperatures compared to at least some other calibrations [e.g. Zaarur et al., 2012], depending on what is meant by 'moderate'. And why preliminary?

We have updated the text to clarify the term "moderate" (20–40°C). We use the term "preliminary" because few data have been published using this calibration and direct comparison of estimates using multiple calibrations is uncommon. However, we are confident that this calibration is the most appropriate for our data and addresses many previous issues with discrepant calibrations (e.g., Zaarur et al., 2013; see Kelson et al.,

2017).

Section 2.2. The introduction and description of the model results is extremely brief which is ok given that these data are previously published (although a few more details would be helpful), but what I did miss was an explanation of why this model was chosen for comparison. Why not one (or the ensemble) of EoMIP simulations?

We did not describe the models in detail because, as noted, each has been previously published. We have added text explaining where further information on each model may be located. Additionally, we have added text explaining why this Eocene model was chosen over the EoMIP ensemble (because the regional model provides better resolution over the GRB domain).

Line 317. It would be useful to elaborate on what is meant by 'isotopic data'. Is it that the clumped isotope results look reasonable, or do you mean calculated $\delta$18Ow look reasonable? If the latter, state what these are in the text in this paragraph (perhaps around lines 323-326).

We have modified the text to clarify that we were referring to clumped isotope data here. Additionally, we include a reference to Table B.1 in the subsequent sentence, which contains the data discussed in the $\delta$18Ow comparison as well.

Lines 363-366. This sounds encouraging except that a $\pm3°$C uncertainty in both WMMT and CMMT results in a seasonality uncertainty of 6°C, given that presumably there may be systematic biases across the reconstructed interval?

We believe that the true error on these estimates is likely much smaller than the reported uncertainty of $\pm3°$C; as explained in the text, we arrive at this uncertainty by doubling the calculated uncertainty (95% CI) for these measurements in order to account for any unforeseen systematic biases in the method or assemblages themselves. Regardless, if we assume that this maximum error is possible, this results in an MART uncertainty of up to 6°C, which still predicts higher (>20°C) MART during the early

Eocene than previous work, as discussed at the end of Section 4.2.

Section 4.2. Somewhere near the beginning of this section please state how EECO and pre-EECO seasonality were calculated. Are the seasonality estimates derived by comparing the mean of (e.g.) all CMMT data to the mean of all WMMT data for that interval? Judging by eye, there seems to be no difference between EECO and pre-peak-EECO MAT and WMMT if all the data are averaged.

We agree that Section 4.2 lacked a description of how these values were calculated, and have added text to explain that the MART estimates for peak and non-peak intervals are derived by averaging all WMMT and MAT (and CMMT) data for each interval (e.g., Table 1). As discussed in Section 4.2, these do produce different estimates that appear to be statistically distinct.

Lines 397-398. It was not clear to me the first time I read this whether the numbers in brackets refer to the modern or Eocene seasonal range, especially on line 397, reword for clarity.

Rearranged for clarity, as per reviewer.

Lines 396-401. As you state earlier, both MAT and WMMT may be cool-biased though, so the agreements and uncertainties discussed here only apply if this is not the case. Consider reemphasising this.

Cool-biasing in the paleobotanical records is often minimal (possibly 2–4°C; Kowalski and Dilcher, 2003), and any effect would likely impact both WMMT and MAT records, resulting in either no or a slight positive change in overall MART. Thus, it is unlikely to impact how well these proxy records and models agree, and any difference would make low MART values even less likely, further strengthening our argument that seasonality was not reduced during the early Eocene. The text of Section 4.2 has been updated to emphasize this, as per the reviewer.

Lines 433-435. This sentence is worded too strongly. There are no reconstructions of

CMMT during the EECO, but moreover the WMMT proxy in the EECO is different to the WMMT proxy in the pre-peak-EECO interval. This latter issue is especially problematic given that the clumped and paleobotanical evidence are not in agreement in the interval for which both are available. I fully sympathise with the authors in that continental climates are difficult to reconstruct, and as I said before, the recognition and discussion of uncertainties in this manuscript is excellent overall. However, I recommend stating these issues more clearly at this point in the text including a more thorough discussion of whether the peak and pre-peak EECO seasonalities are really distinguishable from each other.

While we agree that it would be ideal to have both types of proxy data throughout this interval, the WMMT proxy is not entirely different for these periods, as clumped isotope data exists for both the peak and non-peak parts of the EECO (Figure 4). We fully discuss the relative uncertainties in both types of WMMT estimates (Section 4.1), and while it is possible that the observed trend of expanded MART during the peak EECO may be related to a lack of CMMT data for the interval, the consistency of both types of MART reconstruction (WMMT-CMMT and 2*[WMMT-MAT]) outside of the peak period suggests this trend is real (see response above). We have added text explaining this in greater detail (Section 4.3). Assuming these reconstructed MARTs are robust, statistical tests show that the peak and non-peak periods really are distinct (see comments above; Section 4.2).

Figure 4. Either remove the left half of the peak EECO bar at the bottom of the figure or shade it white so that it is clear that this has not been reconstructed. Please clarify whether the tick marks represent the midpoint between CMMT and WMMT or whether they are placed at the location of the mean MAT estimates.

We have not removed the peak EECO bar as suggested because we argue that while this work does not directly reconstruct CMMT for that period, it is appropriate to reconstruct MART as twice the difference between MAT and WMMT (see response above). Thus, the full range should be represented on the figure as shown. As described in

the figure caption, the vertical tick marks on MART represent the estimated MAT value, which coincidently is very near to the statistical "midpoint".

Supplement. Please given ages with all reconstructions and not just heights, or at the very least state at which heights the peak EECO occurred.

Not all samples have been assigned absolute ages; however, the supplement has been updated as per reviewer to indicate where in the stratigraphy "peak EECO" has previously been interpreted (Hyland and Sheldon, 2013; Hyland et al., 2017).

Line 122 'elements'.

Corrected as per reviewer.